# UV-LED Combined with Small Bioreactor Platform (SBP) for Degradation of 17α-Ethynylestradiol (EE2) at Very Short Hydraulic Retention Time

**DOI:** 10.3390/ma14205960

**Published:** 2021-10-11

**Authors:** Oran Fradkin, Hadas Mamane, Aviv Kaplan, Ofir Menashe, Eyal Kurzbaum, Yifaat Betzalel, Dror Avisar

**Affiliations:** 1School of Mechanical Engineering, Faculty of Engineering, Tel Aviv University, Tel Aviv 69978, Israel; oranfradkin20@gmail.com (O.F.); hadasmg@tauex.tau.ac.il (H.M.); yif.bet@gmail.com (Y.B.); 2The Hydrochemistry Laboratory, The Water Research Center, Porter School of the Environment and Earth Sciences, Faculty of Exact Sciences, Tel Aviv University, Tel Aviv 69978, Israel; avivkaplan@tauex.tau.ac.il; 3Water Industry Engineering Department, Achi Racov Engineering School, Kinneret College on the Sea of Galilee, M.P. Emek Ha’Yarden 15132, Israel; ofirmn@mx.kinneret.ac.il; 4BioCastle Water Technologies Ltd., Tzemach Industries Central Area, Jordan Valley 15105, Israel; 5Shamir Research Institute, University of Haifa, Qatzrin 12900, Israel; eyal3056@gmail.com

**Keywords:** ultraviolet light-emitting diode, Small Bioreactor Platform Capsules, oxidative bioreactor, biomass encapsulation

## Abstract

Degradation of 17α-ethynylestradiol (EE2) and estrogenicity were examined in a novel oxidative bioreactor (OBR) that combines small bioreactor platform (SBP) capsules and UV-LED (ultraviolet light emission diode) simultaneously, using enriched water and secondary effluent. Preliminary experiments examined three UV-LED wavelengths—267, 279, and 286 nm, with (indirect photolysis) and without (direct photolysis) H_2_O_2_. The major degradation wavelength for both direct and indirect photolysis was 279 nm, while the major removal gap for direct vs. indirect degradation was at 267 nm. Reduction of EE2 was observed together with reduction of estrogenicity and mineralization, indicating that the EE2 degradation products are not estrogens. Furthermore, slight mineralization occurred with direct photolysis and more significant mineralization with the indirect process. The physical–biological OBR process showed major improvement over other processes studied here, at a very short hydraulic retention time. The OBR can feasibly replace the advanced oxidation process of UV-LED radiation with catalyst in secondary sedimentation tanks with respect to reduction ratio, and with no residual H_2_O_2_. Further research into this OBR system is warranted, not only for EE2 degradation, but also to determine its capabilities for degrading mixtures of pharmaceuticals and pesticides, both of which have a significant impact on the environment and public health.

## 1. Introduction

The growing scarcity of water resources calls for alternative solutions to producing more valuable water sources. These solutions are based on various strategies for water-saving, water production and wastewater reclamation. The traditional biological-based wastewater treatment is insufficient to remove contaminants of emerging concern (CECs), such as pharmaceuticals, personal care products and hormones, which are mostly persistent compounds that have been detected in water at low concentrations (ng/L to µg/L). The highest level of EE2 has been detected in domestic wastewater was up to 42 ng/L and for surface water was up to 17,112 ng/L [1,2]. Consequently, the effluent from traditional wastewater treatment plants is not adequate for indirect or direct potable reuse, where the reclaimed water is introduced directly into the drinking water system. Therefore, innovative technologies for the removal of CECs from effluent are needed.

The term advanced oxidation process (AOP) refers to ozonation, or a combination of ozonation or UV irradiation with H_2_O_2_ (for indirect vs. direct degradation). Also UV alone as oxidation/photolysis process has the ability to eliminate CECs from wastewater. It is usually applied as a tertiary treatment [3,4,5,6,7,8]. There are two major drawbacks to AOP systems: the formation of unknown degradation products, which can be more toxic and stable than the parent compounds [9,10], and this technology’s high construction and energy costs.

UV wavelengths range between 100 nm and 400 nm, commonly subdivided into vacuum UV (100 to 200 nm), UV-C (200 to 280 nm), UV-B (280 to 315 nm), and UV-A (315 to 400 nm) [11]. UV light disinfection has been common practice for pathogen inactivation in drinking water and wastewater for many years [12]. In addition, UV can promote photolytic and oxidative degradation of CECs in water [8,13,14]. Today, two types of mercury vapor-filled lamps produce germicidal UV irradiation. The first is a low pressure (LP) lamp, monochromatic at about 253.7 nm. The second is a medium pressure (MP) lamp, polychromatic with higher mercury pressures that emits light at multiple peaks from 200 nm and above, including 254 nm. The Minamata Convention on Mercury states the global position regarding the use of mercury products. However, LP and MP mercury lamps were excluded from this convention, as it was claimed that no feasible mercury-free alternative is available [15]. Nevertheless, LEDs could replace them [16]. 

Applications for light-emitting diode (LED) bulbs, which are a popular alternative to fluorescent lighting, have increased in the last decade. As an effective ultraviolet (UV) source, LEDs have significant advantages over mercury vapor lamps. UV-LEDs are physically small, mercury-free, and characterized by fast startup time and long lifespan compared to the traditional mercury vapor lamps [17]. LEDs also emit light at specific wavelengths, allowing for flexible tailoring of UV system designs that combine selected wavelengths to achieve optimal inactivation of various pathogens [18], and mineralization and degradation of persistent contaminants [19,20,21,22]. Being a new technology, UV-LEDs suffer from low power and energy efficiencies at the shorter wavelengths; however, these parameters are constantly being improved [23].

The goal of this study was to evaluate a new treatment approach—the oxidative bioreactor (OBR) system—which was developed in our laboratory. The idea behind this system is to combine biological degradation with UV irradiation, applied simultaneously. The proposed system depends on the benefits and advantages of the small bioreactor platform (SBP, capsules) technology described in US Patent No. 8,673,606, which allows for protection of viable selective biomass by a microfiltration membrane. When the SBP is coated with a semipermeable membrane, penetration of UV light into the capsule medium is prevented, thus providing a protective shield against UV irradiation. The capsules, which work well at short hydraulic retention times (HRTs) typically <6 h, [24,25,26], allow only dissolved molecules to cross the membrane while retaining the microorganisms in the capsule [27,28,29]. This protective barrier allows the microbial culture to survive and prosper under the UV irradiation treatment commonly used in AOPs. Moreover, the SBP may serve as an advanced treatment for municipal wastewater; in our laboratory, SBP-encapsulated cultures of *Rhodococcus zopfii* exhibited faster biodegradation of the steroidal sex hormone 17α-ethynylestradiol (EE2) [3,4,24,29]. Here we examined the degradation efficiency of the OBR and evaluated the system’s potential future implementation as a treatment model for molecules that resist biodegradation, such as pharmaceuticals. The examined OBR system is intended for use as a treatment apparatus in domestic and industrial wastewater-treatment plants for removal of persistent organic contamination.

The potential of combining physical and biological treatments is not a new concept. It was proven a decade ago by Yongming Zhang and co-workers who used an integrated photocatalytic–biological reactor for trichlorophenol degradation [30,31]. Moreover, since direct UV radiation had a strong negative effect on the introduced biomass, they also used a photochemical reaction with visible light and catalyst (TiO_2_). They found that trichlorophenol removal by the coupled photocatalytic–biological reactor was faster than that by photocatalysis or biodegradation alone using a ceramic honeycomb biofilm carrier, or by sequentially coupled photocatalysis and biodegradation. Dandan Zhou and his co-workers used a photolytic circulating-bed bioreactor for phenol treatment. The macroporous carriers were used to support the biofilms encasing the treatment biomass. They used two UV light sources on each side of the bioreactor. The microbes in the biofilm, especially at the surface of the carrier, suffered fatal levels of damage to the cell membranes or DNA, and large amounts became detached because they were exposed to UV for 16 h, indicating the importance of protecting the biomass from UV radiation [32]. A recent review of coupled photocatalysis and biodegradation treatment presents the novelty of this treatment configuration [33]. In reviewing most coupling technologies, those authors concluded that future applications of carriers will also rely on the development of high-strength, bio-based, porous, repeatable 3D carriers with easy-to-implement synthetic solutions. Moreover, to improve the function of microbial communities, the sources of microbial inoculation for emerging pollutants should be optimized for degradation efficiency of the target pollutants. The SBP capsules are applicable as they provide a solid membrane, allowing for long-term (on the order of two months) implementation with the bioreactors and enabling the integration of catalytic particles [34]. Moreover, in the introduced bacterial culture, the bacteria are suspended in a confined aquatic medium that allows for relatively high rates of contaminant movement and passage. The objective was to examine a novel technique that combines biological (SBP) and physical (UV-LED) degradation of EE2 in a combined reactor (OBR).

## 2. Materials and Methods 

### 2.1. UV Light 

#### 2.1.1. Batch Setup 

Prior to optimal construction of the OBR, two laboratory batch reactors were used at different wavelengths to determine maximal EE2 removal. Consequently, two LED systems were purchased that included, approximately, the chosen wavelengths but with a higher magnitude of irradiance for the OBR reactor. The “rectangular system” included LEDs with peak emission wavelengths at 265 nm and 285 nm (six LEDs for each wavelength divided between two LED circuit boards), with an overall area of 15.7 cm × 11.5 cm (lenght × width). The 265 nm and 285 nm UV-LEDs displayed peak wavelength emissions of 267.2 nm and 285.8 nm with full width at half maximum (FWHM) bandwidths of 12.3 nm and 13.5 nm, and incident irradiances (mW/cm^2^) of LED_265_ = 0.164 and LED_285_ = 0.407, respectively.

The “circular system” had LEDs with peak emission wavelengths at 275 nm (nine LEDs for each wavelength and a collimating chamber) with an overall area of 7.5 cm (diam.). The 275 nm UV-LED exhibited peak wavelength emission of 278.8 nm, with FWHM bandwidth of 20 nm and incident irradiance (mW/cm^2^) of 0.426 (Appendix A Appendix A).

#### Optimization of Wavelengths

For wavelength adjustments, photolysis experiments were conducted in triplicate with both rectangular (Appendix A Appendix A) and circular (Appendix A Appendix A) UV-LED systems. The solution was composed of EE2 (0.5 mg/L) in phosphate buffer (pH 7.5, 10 mM) with the option of adding 10 mg/L hydrogen peroxide (H_2_O_2_) in a 105-mL laboratory beaker at 25 °C. The solution was stirred with a magnetic stirrer and 1-mL aliquots were collected at predetermined time intervals for analysis of EE2 degradation (HPLC-UV).

#### 2.1.2. OBR

The OBR system (Figure 1) had LEDs with peak emission wavelengths at 275/265 nm, 20 LEDs for each wavelength, with an overall area of 5.6 cm × 4 cm (lenght × width). The 265 nm and 275 nm UV-LEDs exhibited peak wavelength emissions of 267.7 nm and 277.4 nm, respectively, with FWHM bandwidths of 10.8 nm and 10.9 nm, respectively, and with incident irradiances (mW/cm^2^) LED_265_ = 3.36 and LED_275_ = 6.02 (Appendix A Appendix A). The OBR and the controls were tested as a batches (each one with), in a total volume of 3 L at 25–30 °C and stirred with a magnetic stirrer. The circulation was done by the magnetic stirrer and was examined up to 24 h. Each experiment was performed in triplicate. 

#### Total Organic Carbon (TOC) and Estrogenicity Experiments 

Photolysis experiments were conducted with UV-LEDs in the OBR without SBP capsules (Figure 1B). EE2 solution at the highest concentration (8.5 mg/L), with or without 50 mg/L H_2_O_2_. The high concentration was taken due to the lack of the sensitivity of the TOC measuring instrument. Then 1 mL of solution was taken for estrogenicity analysis (by yeast estrogen screen (YES)) and 15 mL was taken for TOC analysis.

#### OBR Experiments

The OBR is an innovative technology, which combines UV-LED with SBP-protected biomass. First, we evaluated the influence on secondary effluents, which were taken from a domestic wastewater-treatment plant (Shafdan, Petah Tikva, Israel) spiked with 0.5 mg/L EE2. We compared four different processes: (i) LEDs—physical process (Figure 1B); (ii) SBP—biological process (Figure 1C); (iii) OBR which combines, simultaneously, physical, and biological processes (LEDs with SBP) (Figure 1A); (iv) indirect AOP, a physical–chemical process with addition of 10 mg/L H_2_O_2_. Then, we repeated the experiments with water enriched with 0.5% LB, 10 mg/L sodium nitrate, 0.5 mg/L ferric phosphate, and 0.5 mg/L EE2. For each experiment, 3-mL samples were taken at predetermined time intervals to examine the degradation of EE2 using HPLC-UV (Hydrochemistry Laboratory at Tel Aviv University, Tel Aviv-Yafo, Israel) analysis.

### 2.2. Encapsulation of Bacterial Cultures in SBP Capsules

The main reason for using the SBP capsules was to preserve and protect the bacteria, while simultaneously exposing them to UV irradiance. This enabled the examination of an innovative OBR as an AOP that combines physical and biological processes (UV-LED together with SBP(-encapsulated bacteria)). An EE2-degrading *Rhodococcus zopfii* strain (ATCC 51349) was purchased from the ATCC Culture Bank (Manassas, VA, USA), and the SBP capsule “Research Kit” (Catalogue number AC-20) was obtained from BioCastle Water Technologies Ltd. (Samakh, Israel). To encapsulate the specific bacteria, a laboratory activation procedure was developed. A volume of 1.2–1.5 mL of the culture suspension containing *R. zopfii* was injected into 45 sterile SBP capsules, and the injection hole was then sealed with cellulose acetate polymer, provided with the research kit. The bacterial filling procedure is presented at [35]). Before using the cultured SBP, the capsules were incubated for 48–72 h in a solution containing 2% Lennox broth (LB) (Sigma–Aldrich, Jerusalem, Israel). 

In preliminary experiments, proof of the efficacy of SBP capsules in providing protection of a sustained biomass under UV irradiation was obtained. First, the suspended and SBP-encapsulated bacteria were exposed to UV irradiation. There was no change in the concentration of bacteria in the suspended or encapsulated states as long as the UV lamp was off (0–100 min). When the UV lamp was turned on (100–220 min), the suspended bacterial concentration decreased rapidly (from 1.10 × 10^7^ to 3.03 × 10^3^ CFU/mL), whereas no change was observed in the SBP-encapsulated culture. These results demonstrated the resistance of the SBP-encapsulated microorganisms to over 100 min exposure to UV radiation, and showed that the capsules provide a sufficient protective barrier for the encapsulated bacterial culture (Appendix A Appendix A). These preliminary results provided our proof of concept for OBR treatment development and use.

### 2.3. Experimental Stock Solution

EE2 stock solution (1050 mg/L) was prepared by mixing 50% acetonitrile and 50% DI (deionized) water (*v*/*v*), and was diluted directly in the experimental flasks.

In the TOC experiment, EE2 was dissolved directly to OBR (Figure 1B) in order not to interfere with the TOC and estrogenicity measurements.

### 2.4. Analytical Measurements

#### 2.4.1. HPLC-UV

EE2 was analyzed by HPLC-UV (Agilent 1100 Series instrument, Anaheim, CA, USA) using a Kinetex 2.6 µm EVO C18 column, 100 × 3 mm, with mobile-phase composition of 55% water and 45% acetonitrile (*v*/*v*), in isocratic mode at a flow rate of 0.6 mL/min. Injection volume was 90 µL. Initial EE2 concentration and at monitoring time were designated EE2_0_ and EE2_t_, respectively. The calibration curve was between 0.005 to 17.5 mg/L.

#### 2.4.2. TOC

TOC method was used to measure the degree of mineralization. TOC was measured by Aurora TOC analyzer (O.I. Analytical, Lake Success, NY, USA). The instrument measures TOC by acidifying and oxidizing the carbon in the sample solution into CO_2_ and calculating its concentration using a calibrated infrared reading detector.

#### 2.4.3. Colorimetric Estimation of H_2_O_2_ Using Strips

The amount of introduced and consumed H_2_O_2_ was estimated by peroxide test strips (Mquant, Merck, Darmstadt, Germany). The aqueous suspension was stirred at regular intervals, a 2-mL aliquot was passed through a 0.45 µm filter, and the color change of a strip dipped in the supernatant was observed, the ranges were 0, 1, 3, 10, 30, 100 and 0, 0.5, 2, 5, 10, 25 mg/L.

#### 2.4.4. UV-LED Emission Spectra

UV-LED emission spectra were measured in an Ocean Optics USB4000 spectroradiometer equipped with a cosine corrector (Ocean Optics OCF-104447, EOS-A1241604, Orlando, FL, USA).

#### 2.4.5. Estrogenicity Assay—YES

YES was performed to verify the decrease in estrogenic activity in the treated water. This bioassay is a useful tool for evaluating the effectiveness of AOPs and biotreatments, and to determine suitable operating conditions. Yeast cells were purchased from Xenometrix (Allschwil, Switzerland). YES assay [36] was performed with modifications [37]. Briefly, 100-μL dilutions of EE2 standards (0–10 nM) and samples from the experiment were added to sterile 2-mL Eppendorf tubes containing 300 μL diluted yeast solution with an optical density at 630 nm (OD_630_) of 0.065. Tubes were covered with Kimwipes (Kimtech Science, Roswell, GA, USA) and incubated for 3 days at 30 °C with gentle shaking (100 rpm). After the incubation, 50 μL of Z-buffer was added to each tube. The enzymatic reaction was initiated by adding 400 μL assay buffer (containing 1 mg/mL ortho-nitrophenyl-β-d-galactopyranoside and Z-buffer supplemented with β-mercaptoethanol and 10% *v*/*v* sodium dodecyl sulfate lysis buffer). After 20 min of incubation at 30 °C and stirring at 100 rpm, 200 μL of 1 M sodium carbonate was added to stop the enzymatic reaction. The tubes were centrifuged at 3000 rpm for 10 min, and 100 μL supernatant from each tube was transferred to a 96-well microtiter plate to determine OD_405_ and OD_630_. The calculation of OD_405_–OD_630_ gave the relative estrogenic activity of each sample.

## 3. Results and Discussion

### 3.1. Photolysis—Wavelength Selection

In this experiment, we evaluated EE2 degradation at three UV wavelengths—267.2, 285.8, and 278.8 nm—without (direct photolysis) and with (indirect photolysis) at 10 mg/L H_2_O_2_ in the rectangular and circular UV-LED system, respectively.

Under direct photolysis, with UV radiation of 267.2, 278.8 and 285.8 nm, EE2 degradation was 14.7%, 35.3%, and 26.2%, respectively, after 60 min. Under indirect photolysis, UV radiation at 278.8 nm gave the highest EE2 degradation (92.7%), showing approximately 10% higher efficiency than the 285.8 and 267.2 nm wavelengths (83.2%) (Figure 2).

After calculation, the largest removal gap (68%) between direct and indirect degradation was obtained at 267.2 nm. Therefore, wavelengths of 278.8 and 267.2 nm were selected for further experiments.

### 3.2. Relationship between EE2 Degradation, Estrogenicity, and Mineralization

Removal of EE2 under direct photolysis (at 267.7 nm and 277.4 nm), was 50%, 73%, and 98% at 4, 8, and 24 h, respectively. Removal of EE2 under AOP photolysis (with 50 mg/L H_2_O_2_) was 50%, 90%, 99%, and 99.9% after 0.33, 1, 2 and 3 h, respectively. The results obtained for estrogenicity reduction by direct photolysis showed approximate decreases of 23%, 70%, and 96% after 1, 8, and 24 h, respectively. The results of the AOP photolysis showed approximate decreases of 48%, 98%, and 99.9% after 0.33, 2, and 24 h, respectively (Figure 3).

A high correlation was found between EE2 degradation and reduction of estrogenicity, indicating that the obtained EE2 degradation products are not estrogenic (under both direct and indirect photolysis). In addition, the TOC results obtained with the direct UV-LED radiation showed only slight mineralization (3%). Upon addition of 50 mg/L H_2_O_2_, degradation of 65% of the TOC was observed after 24 h; approximate mineralization of 3%, 10.5%, and 18.5% was observed at 3, 5, and 7 h, respectively (Figure 3). This linear trend was derived from almost full EE2 degradation (starting at 3 h), and the significant decrease in estrogenicity, which may indicate that most of the irradiance energy goes to high concentrations of OH radicals, which then concludes the mineralization process.

### 3.3. OBR as a Substitute for Physical–Chemical Treatment of Effluent 

An innovative alternative based on physical–biological processes, the OBR, was tested. The OBR combines UV-LED irradiation with SBP-protected biomass to optimize two major mechanisms: biodegradation and oxidation. The test medium was secondary effluent enriched with EE2 (Figure 4). 

The UV-LEDs combined with (10 mg/L) H_2_O_2_ AOP showed better EE2 degradation performance only after 4 h of incubation (Figure 4). However, the main disadvantage to using this process to degrade EE2 is the residual H_2_O_2_: 10, 8, 7, 5, 0.2 mg/L residual H_2_O_2_ was measured after 0, 2, 4, 8, and 72 h, respectively. Moreover, the EE2 remained stable after the irradiation was stopped, and the H_2_O_2_ measurement indicated that H_2_O_2_ alone (without irradiation) does not degrade EE2. The H_2_O_2_ residuals might attach to other molecules and create more hormonal secondary metabolites. The combined biological and physical treatment results indicated that up to 43 h, the OBR gives the best performance, even better than the AOP. This indicated that the OBR technology can be implemented in secondary sedimentation tanks that in most cases involve 4–6 h of HRT. Furthermore, instead of two separate beakers, which will double the time, this technology allows us to consolidate the process into one beaker at the same time. Still, EE2 degradation after 4–6 h of incubation amounted to approximately 26–32%, which is far from that expected for tertiary treatment which requires at least 70% contaminant reduction. 

The OBR system was also evaluated at very long HRT (24 h) to better understand whether the HRT is the only restrictive factor for treatment efficacy. After 24 h, EE2 degradation was 91%, 72%, and 48% for the AOP, UV-LEDs and the OBR, respectively (Appendix A Appendix A). Lower efficacy of OBR with respect to AOP and UV-LED alone can be explained by two factors: (i) medium turbidity increases with time due to membrane release of fragments of cellulose acetate into the medium, leading to a reduction in UV efficacy; (ii) a possible shading effect by the capsules, which might also reduce radiation efficiency. Both factors might reduce the potential efficacy of the OBR: the turbidity dramatically changes the results over long HRTs, and shading reduces photolysis efficacy, as some of the photons are absorbed by the capsules and the cage. In a continuous treatment system, these factors are expected to be somewhat reduced. For example, fragment release and turbidity increase are limited to small water volumes, as demonstrated in the laboratory.

### 3.4. Comparison of Two Different Media in the Different Processes

To strengthen the initial results, further examination of the physical–biological processes in the OBR was conducted. In these experiments, the different water-treatment approaches were evaluated in two types of medium for EE2 degradation. 

Nutrient-enriched medium solution is relatively similar to secondary effluent in terms of absorbance (Appendix A Appendix A). The results of the combined OBR process for the enriched-water solutions indicated an advantage for EE2 degradation compared to the physical and biological processes alone. With the OBR system, degradation of 23%, 34%, 46%, and 54% at 1, 2, 4, and 6 h, respectively, was obtained, whereas UV-LED irradiation alone degraded 14%, 23%, 36%, and 45% at the same respective time points, and the biological process alone led to only 15%, 22%, 30%, and 33% degradation, respectively. This additive effect (up to 60% benefit) was obtained only under very short HRT (vsHRT), i.e., less than 6 h (Figure 5). Up to 6 h, the observed kinetic rates were 0.045, 0.03, and 0.028 mg/h for the OBR, physical, and biological approaches, respectively.

However, after 24 h, the obtained results (80%, 75%, and 46% for the UV-LED, combined process and SBP capsules, respectively) only showed an advantage for the physical process, while the combined process resulted in a lower degradation rate, as described in Section 3.3 (Figure 5). The kinetic rate for the time interval of 6–24 h was 0.01, 0.006, and 0.004 mg/h for the physical, OBR, and biological treatment, respectively. 

As already shown, the effluent medium demonstrated a trend similar to that observed with the enriched-water medium. EE2 removal was 18%, 32%, and 48% for the combined process, 8%, 23%, and 72% for the LED process, and 14%, 23%, and 25% for the biological process, at 2, 6, and 24 h, respectively (Figure 5). For the first 6 h, the observed kinetic rates were 0.027, 0.019, and 0.019 mg/h for OBR, physical, and biological treatment, respectively. In the effluent medium, the kinetic rates for the 6–24 h interval were 0.014, 0.004, and 0.0006 mg/h for physical, OBR and biological treatment, respectively. During vsHRT (approximately 6 h), the combined OBR process reduced EE2 by 32%, while the physical process only reduced it by 23%, showing an advantage of the combined OBR process for short time spans, as seen with the enriched-water medium. 

## 4. Summary and Conclusions

The main objective was to evaluate the OBR as an innovative technology that couples a biological treatment using SBP-protected biomass and a physical treatment using UV-LEDs to degrade EE2. For secondary effluent, which is the major source for treatment, an advantage was found for the physical–biological process at vsHRT, with the highest benefit (>60%) this advantage occurs up to 4 h, indicating good potential for use of the OBR in a secondary clarifier. The advantages at vsHRT (up to 6 h) will allow us to implement the OBR system in secondary sedimentation tanks, which, in most cases, have 4–6 h of HRT. The results of this study reveal that under the right conditions, UV-LEDs adjustment, and time span up to 4 h, an additive treatment effect, physical and biological, can be achieved in the same tank simultaneously by the OBR treatment. In addition, it is possible to reduce the expected negative effect of the SBP, which reduces the physical treatment efficacy. This study is the first to examine and evaluate the effect of treatment coupling (physical and biological), using an OBR configuration, on a steroidal sex hormone (EE2), which can be modified for the treatment of additional micropollutants. Improving the OBR’s EE2 treatment efficacy will provide a technology that is suitable for implementation in secondary sedimentation tanks in compliance with the process HRT. However, after 24 h of the combined treatment process, we noticed that EE2 reduction was reduced to 48%; UV-LED irradiation (physical) alone reduced the EE2 by 72%, demonstrating better performance. Thus, further improvements of the OBR’s design are required, and research should continue to explore this system, not only for EE2 degradation, but also for the degradation of mixtures of pharmaceuticals and pesticides, which are considered to have a significant impact on the environment and public health. It should be noted that previous studies combining biological and catalyst treatments used visible light combined with a catalyst, to avoid radiation damage to the introduced biomass. Here, the SBP-encapsulated biomass was not adversely affected by the UV light and therefore, the technology presents a significant advantage for exploring the treatment potential of UV radiation. This is the first demonstration of use of the OBR on EE2 biodegradation and further study is warranted. 

## Figures and Tables

**Figure 1 materials-14-05960-f001:**
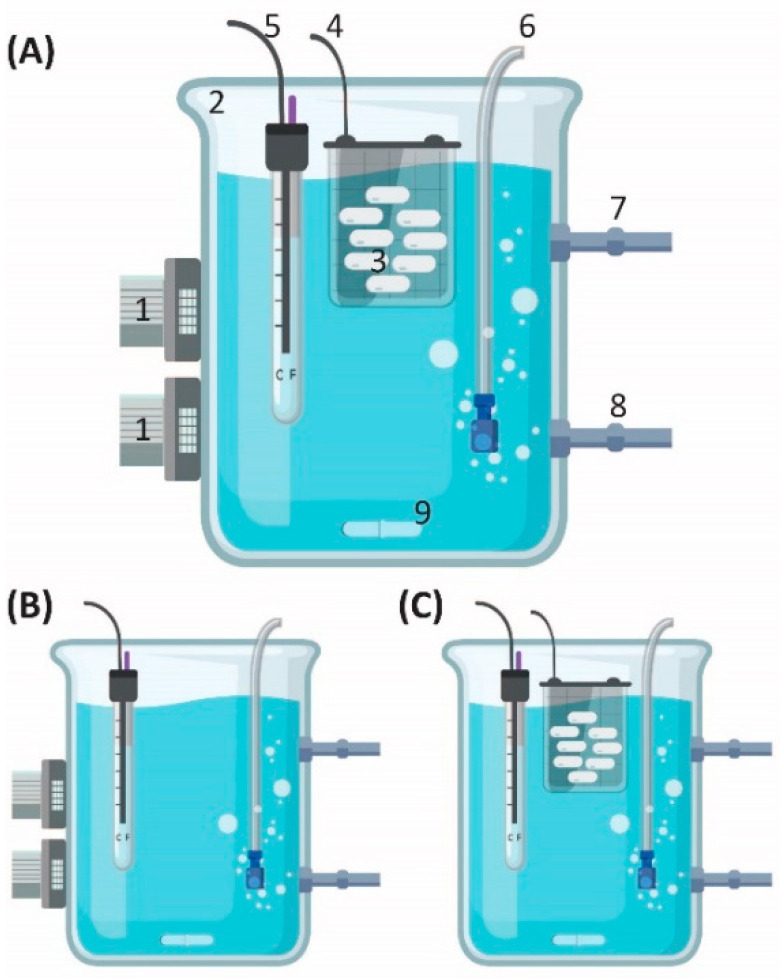
The OBR. (**A**) 1. UV-LEDs at two wavelengths, 267.7 nm and 277.4 nm; 2. quartz beaker of maximum 3.8 L volume (diam. 135 mm × height 280 mm); 3. SBP capsules; 4. beaker-shaped grid to hold the SBP capsules; 5. cool/warm water (thermostat); 6. air diffuser; 7. and 8. inflow and outflow stream to new OBR; 9. stirrer. (**B**) System without SBP capsules as a control. (**C**) System without UV-LED as a control.

**Figure 2 materials-14-05960-f002:**
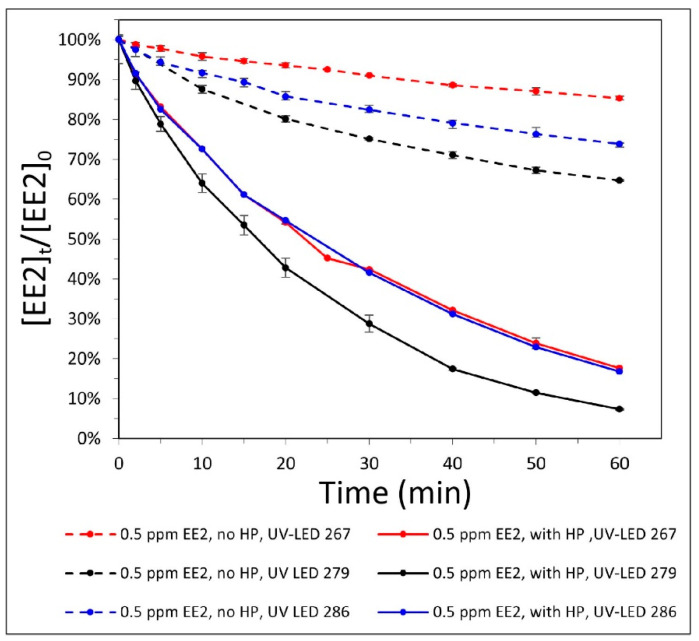
Degradation of 0.5 mg/L EE2 by direct UV-LED or indirect AOP with 10 mg/L H_2_O_2_ (HP), using various LED wavelengths, versus time.

**Figure 3 materials-14-05960-f003:**
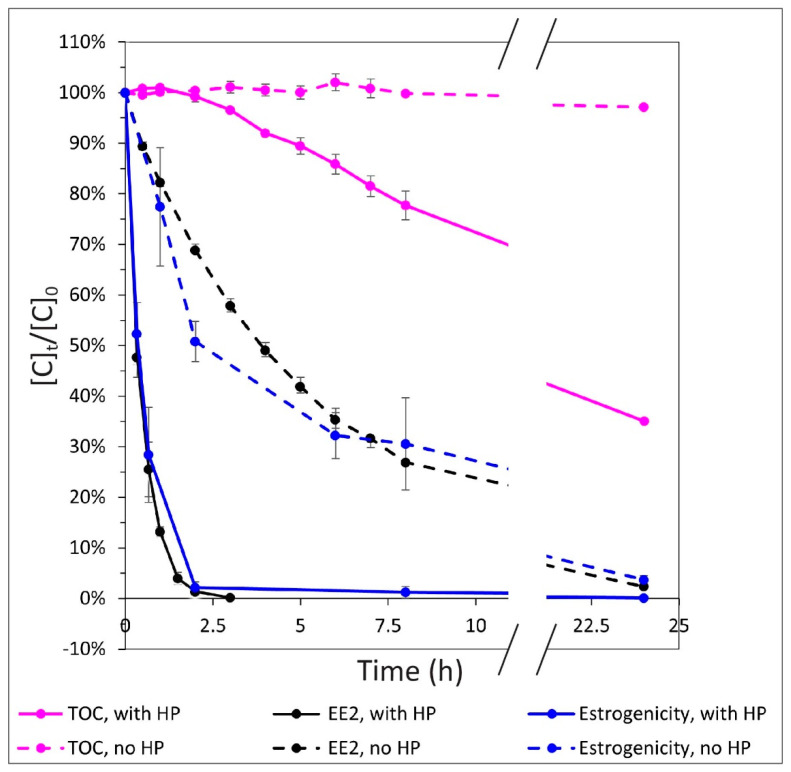
Mineralization, estrogenicity, and EE2 (8.5 mg/L) degradation by direct UV-LED or AOP photolysis (50 mg/L H_2_O_2_ [HP]), using two LED wavelengths (267.7 nm and 277.4 nm), versus time.

**Figure 4 materials-14-05960-f004:**
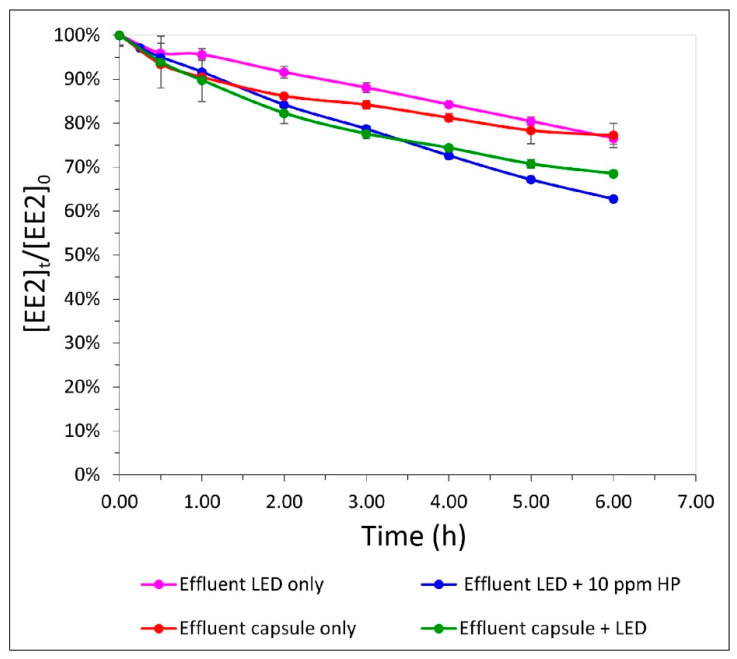
Comparison of EE2 hormone (0.5 mg/L) degradation in 3 L of effluent wastewater, using different processes: UV-LED at wavelengths 267.7 nm and 277.4 nm, SBP, UV-LEDs with SBP, and UV-LEDs with 10 mg/L H_2_O_2_ (HP) versus time.

**Figure 5 materials-14-05960-f005:**
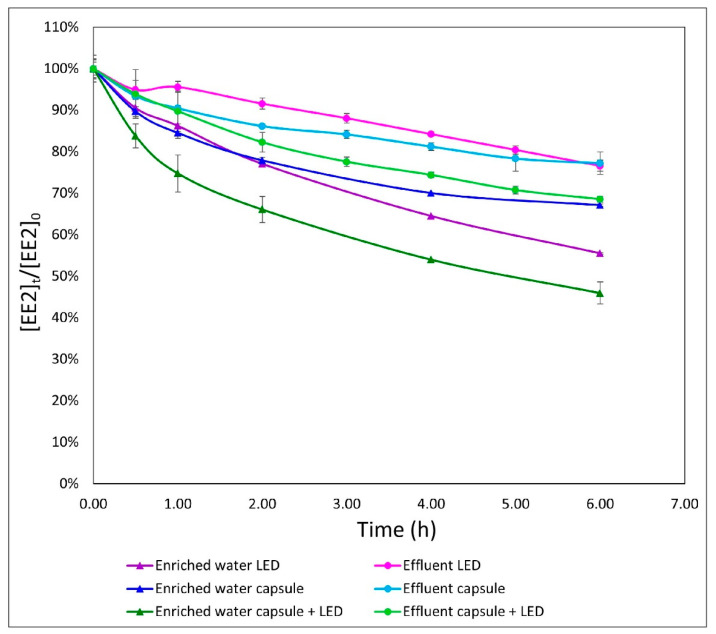
Comparison of two different media: enriched water and effluent wastewater, and three different processes: UV-LED (wavelengths 267.7 nm and 277.4 nm), SBP capsule (encapsulating R. zopfii culture) and both together, in a volume of 3 L with EE2 (0.5 mg/L) versus time.

## Data Availability

Data is contained within the article.

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
