# Peer review of "UV-LED Combined with Small Bioreactor Platform (SBP) for Degradation of 17α-Ethynylestradiol (EE2) at Very Short Hydraulic Retention Time"

_materials, 2021, doi:10.3390/ma14205960_

Round 1
Reviewer 1 Report
This is an interesting report for development of novel oxidative bioreactor that combines physical (UV-LED) and biological (bacteria encapsulated in small bioreactor platform) treatment of estrogen (Ethinylestradiol) enriched water. Combination of physical and biological treatments of wastewater for degradation of pollutants is very complicated task. This study proposed an innovative and cost effective system with high efficiency to degrade estrogen in a short hydraulic retention time. The topic is interesting and important. In general, the experiments were well organized and the results are reliable.
The only weak point of the paper is the presentation. The M&M section have to be revised. The “2.1. UV light characteristics” is not a part of M&M – this is a theoretical description of UV mercury vapor-filled lamps which are not used in this study. On the other hand, there is no characterization of used UV-LED sources.
The numeration of figures, cited in the text, is complete mess.
For example
l. 167 - (Fig. 1) and (Fig. 2) – the context do not correspond with the content of the figures;
l.173-190 figures 3A, B and C does not exist in the manuscript;
l.184 – there is no figure 6 in the paper.
Author Response
Reviewer 1
The only weak point of the paper is the presentation. The M&M section have to be revised. The “2.1. UV light characteristics” is not a part of M&M – this is a theoretical description of UV mercury vapor-filled lamps which are not used in this study. On the other hand, there is no characterization of used UV-LED sources.
Answer: Thanks for the suggestion. The “2.1. UV light characteristics” was removed from the M&M and only described in the introduction as an old technology and why we chose to use UV-LEDs. The UV light and UV-LED characterization is described in lines 51-72 in the paper. In addition, the paper was checked again, and figures numbers were corrected.

Reviewer 2 Report
The objective of the paper was to evaluate the OBR as an innovative technology that couples a biological treatment using SBP-protected biomass and a physical treatment using UV- LEDs to degrade EE2. Results of the study may have important application in the field of wastewater treatment. Authors may wish to consider the following in revision of their manuscript.
- Please comment what pre treatment was carried out for wastewater prior to the proposed treatment used in the study.
- Synthetic wastewater was used in the study. Please comment on organic and inorganic pollutants present in actual wastewater will have on the treatment performance of proposed treatment system.
- Please provide details of bio treatment unit used in the proposed treatment system. How many SBP capsules in each reactor.
- Please provide information of bacterial concentration, for both attached growth and suspended growth, present in treatment unit.
- What is F/M ratio, sludge age used in the treatment system.
- Please provide characteristic of feed wastewater and treated effluent.
- Please provide data regarding TOC removal achieved by different treatment schemes.
- Please comment on the limitation of proposed treatment system.
- Bench scale unit was used in the study. Please comment on the scale up factor to full scale operation.
Author Response
Reviewer 2
The objective of the paper was to evaluate the OBR as an innovative technology that couples a biological treatment using SBP-protected biomass and a physical treatment using UV- LEDs to degrade EE2. Results of the study may have important application in the field of wastewater treatment. Authors may wish to consider the following in revision of their manuscript.
- Please comment what pre treatment was carried out for wastewater prior to the proposed treatment used in the study.
Answer: In this study the most polluted water was effluent water. The effluent was taken from the Shafdan (Israel WWTP). The pre-treatments were: Pre-treatment coarse filtration by bar screens and removal of sand and oil in grit chambers. Primary treatment suspended solids were removed by gravitational sedimentation. Secondary treatment is a biological process intended to treat the dissolved organic materials in the sewage and secondary sedimentation tank to remove the bacteria/colonies. These processes were not done by us. You can see more information in the link https://www.igudan.org.il/home-en/shafdan-wastewater-treatment.
- Synthetic wastewater was used in the study. Please comment on organic and inorganic pollutants present in actual wastewater will have on the treatment performance of proposed treatment system.
Answer: Synthetic wastewater was not used in the study (only effluent was examined). However, I went through the paper to correct misunderstandings about the type of water.
- Please provide details of bio treatment unit used in the proposed treatment system. How many SBP capsules in each reactor.
Answer: In section 2.2 there is data on the SBP and emphasize that in this study we use 45 capsules.
- Please provide information of bacterial concentration, for both attached growth and suspended growth, present in treatment unit.
Answer: Please see reference number 3: “Eleven days after the activation point, where the SBP-encapsulated culture was incubated in MSM medium enriched with EE2, it presented large filamentous colonies (50–1000 μm), which are similar to the structure of colonies that were observed in the suspended state of the culture. Moreover, it seems that the culture has three growth states in the internal medium of the capsule: Filamentous colonies with extensive branching and hyphal growth; single irregular rod-shaped filament units; and single Gram-positive cocci cells. The encapsulated R. zopfii culture formed a floc particles structure and an advanced stage of fragmentation into coccoid elements was also observed. The inner suspension of the SBP capsule contained a mean of 2.88 × 108 CFU/mL R. zopfii at the end of the capsules activation stage.”
- What is F/M ratio, sludge age used in the treatment system.
Answer: Since we did not use wastewater, we believe this information may confuse the reader.
- Please provide characteristic of feed wastewater and treated effluent.
Answer: The feed effluents were mostly urban (Shafdan). In addition, a table of values is attached:
- Please provide data regarding TOC removal achieved by different treatment schemes.
Answer: Please see my answer to questions #1 and #5.
- Please comment on the limitation of proposed treatment system.
Answer: Please see lines 367-370 in the paper.
- Bench scale unit was used in the study. Please comment on the scale up factor to full scale operation.
Answer: I agree that this is a very important parameter. However, future research will examine the OBR in the form of a flow reactor, and will give this parameter.
Another answer: It is difficult to give a final and accurate ratio because the number of capsules in this study was not optimized and we used a batch. From previous paper we can estimate this parameter for full scale operation: 1.25 SBP capsule for a cubic meter per day (https://doi.org/10.1080/09593330.2015.1121293). For our UV-LED the EED (Electrical Energy Dose) is 172 [kWh/m3].

Reviewer 3 Report
The current study was to evaluate the OBR as an innovative technology that couples 354 a biological treatment using SBP-protected biomass and a physical treatment using UV-355 LEDs to degrade EE2. It is a decent manuscript and well organized. But there are still some issues to address. I suggest a minor revision.
1. For Introduction, add the description on the target of your manuscript.
2. Materials and methods, add a section of statistical analyses.
3. Materials and methods, how could the author realize QA/QC.
4. Results and discussion, more discussion on the results should be added. This section is not just a description on the results.
5. Please highlight the innovation of the study in abstract, introduction, and conclusion.
6. Please add significance analyses to Figure 2-5.
7. There are many long sentences. Short sentences are suggested. I recommend the author to change long sentences to short one. Please check and revise throughout the manuscript.
Author Response
Reviewer 3
The current study was to evaluate the OBR as an innovative technology that couples 354 a biological treatment using SBP-protected biomass and a physical treatment using UV-355 LEDs to degrade EE2. It is a decent manuscript and well organized. But there are still some issues to address. I suggest a minor revision.
- For Introduction, add the description on the target of your manuscript.
Answer: Thanks for the suggestion. The description on the target of the manuscript was added in the end of the introduction (lines 117-119).
- Materials and methods, add a section of statistical analyses.
Answer: The M&M section was revised and reorganized, including adding sentences regarding the statistical analyses in the existing chapters.
- Materials and methods, how could the author realize QA/QC.
Answer: The same things were done as described in the answer to the previous question.
- Results and discussion, more discussion on the results should be added. This section is not just a description on the results.
Answer: Indeed, further discussion could be better. However, to avoid clutter and confusion, we made only small changes.
- Please highlight the innovation of the study in abstract, introduction, and conclusion.
Answer: Thanks for the suggestion. The highlights of the innovation were added in lines 117-119 and line 426.
- Please add significance analyses to Figure 2-5.
Answer: There is an analysis of results before and after the figure, small and new things are emphasized. Repetition of information that can be seen in the figure was not added so as not to confuse readers.
- There are many long sentences. Short sentences are suggested. I recommend the author to change long sentences to short one. Please check and revise throughout the manuscript.
Answer: Thanks for your attention. I did my best to make these changes where it were possible.

Reviewer 4 Report
Dear authors,
I enjoyed your interesting work. In the attached file, along with minor issues, you can find my question, with comments and suggestions.
Best regards

Author Response
Reviewer 4
Answer: Thank you very much for your time, attention, and thorough checking. I went over your comments. Please see the PDF and my reference to some of your comments.

Round 2
Reviewer 1 Report
The quality of manuscript was significantly improved in the revised version. The materials and methods now are clearly presented and described in details.
Figures are numbered and cited correctly. I have no additional comments.
Author Response
The reviewer did not ask for further corrections!
Reviewer 4 Report
Dear authors,
thank you for your replies to my previous comments.
In the attached file you can find few further comments and suggestions
Best regards

Author Response
The answers are in the attached file
